# Geospatial Management and Analysis of Microstructural Data from San Andreas Fault Observatory at Depth (SAFOD) Core Samples

**Elliott M. Holmes \*, Andrea E. Gaughan** 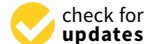 **, Donald J. Biddle, Forrest R. Stevens** 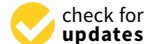 **and Jafar Hadizadeh**

Department of Geography and Geosciences, University of Louisville, Louisville, KY 40208, USA;
ae.gaughan@louisville.edu (A.E.G.); dj.biddle@louisville.edu (D.J.B.); forrest.stevens@louisville.edu (F.R.S.);
jafar.hadizadeh@louisville.edu (J.H.)
**\*** Correspondence: emholm02@louisville.edu; Tel.: +1-502-718-3862

**Abstract:** Core samples obtained from scientific drilling could provide large volumes of direct microstructural and compositional data, but generating results via the traditional treatment of such data is often time-consuming and inefficient. Unifying microstructural data within a spatially referenced Geographic Information System (GIS) environment provides an opportunity to readily locate, visualize, correlate, and apply remote sensing techniques to the data. Using 26 core billet samples from the San Andreas Fault Observatory at Depth (SAFOD), this study developed GIS-based procedures for: 1. Spatially referenced visualization and storage of various microstructural data from core billets; 2. 3D modeling of billets and thin section positions within each billet, which serve as a digital record after irreversible fragmentation of the physical billets; and 3. Vector feature creation and unsupervised classification of a multi-generation calcite vein network from cathodluminescence (CL) imagery. Building on existing work which is predominantly limited to the 2D space of single thin sections, our results indicate that a GIS can facilitate spatial treatment of data even at centimeter to nanometer scales, but also revealed challenges involving intensive 3D representations and complex matrix transformations required to create geographically translated forms of the within-billet coordinate systems, which are suggested for consideration in future studies.

**Keywords:** Geographic Information Systems (GIS); remote sensing; structural geology; 3D visualization; spatial analyses

## 1. Introduction

In recent decades, scientific drilling activities aimed at solid earth research such as tectonic deformation, heat flow, and earthquakes have been on the increase. By examining geophysical logs recorded during drilling and cored rock exhumed from the boreholes thereafter, the scientific community has gained new insight on Earth's subsurface processes and structures. Active fault zone drilling has informed critical advancements in our understanding of fault system dynamics and composition, and how those factors then coalesce to influence seismic hazards experienced by humans at the surface. Structural geologists often utilize traditional analytical techniques such as X-ray diffraction (XRD), cathodoluminescence (CL or SEM-CL), electron backscatter diffraction (EBSD), and optical and electron microscope imaging to gather data from drill cores [1]. Many applications require a combination of several approaches and, though these established techniques generate large volumes of reliable measurements, deriving results via the traditional, piecemeal treatment of the data is often a time-consuming and inefficient process. Geophysical instrumentation and core samples thus allow direct access and observation of the fault zones that is unattainable solely through exhumed fault rocks or historical analysis of seismic events, but the ability to examine spatial relationships and understand multi-scalar subsurface

processes is potentially limited without novel geospatial techniques for integrating various 2D and 3D data sources [2–4].

Unifying core-based data in a Geographic Information System (GIS) allows researchers to locate, visualize, correlate, and explore microstructural characteristics in a streamlined interface. While GIS is most often utilized to manage georeferenced data that span scales of meters to kilometers, it provides robust database management and analysis structures that facilitate spatially explicit treatment of data regardless of its type or scale [5]. As such, in situ microstructural data collected at the nanometer-millimeter scale also fundamentally contain spatial information that can be efficiently managed and analyzed with GIS and remote sensing techniques. Recognizing the value of applying geospatial techniques to microstructural geology allows researchers to maximize the potential of their data, reduce the time needed for spatial analyses of data collected via traditional analytical techniques, and address interdisciplinary questions that were previously challenging.

In the 'traditional' literature, GIS and remote sensing techniques are commonly leveraged in support of community- to global-scale inquiries about anthropogenic climate change, socio-economic conditions, crime distribution, transportation networks, or ecological systems dynamics [6–9]. Studies typically span scales of meters to kilometers and may integrate multiple sources of remotely sensed aerial imagery and vector data acquired with GNSS-enabled devices. One unifying aspect among these applications is the importance of the coordinate space that defines the study area; in other terms, the spatial context of research results is often of equal importance to, or inseparable from, the research itself. In studies involving the remote sensing of land cover, for example, both questions and answers regarding dynamic land systems processes are strongly informed by the geophysical context of the study area, the spatial and temporal resolution of available data, and the computational resources available to process the data. The spatial distribution of patterns and features at the micro-scale has also prompted geologists to explore contemporary GIS techniques for managing microstructural data.

More recently, there is a growing number of studies in the geologic literature regarding the potential applications of GIS frameworks to visualize and analyze microstructural data. Previous work demonstrates the viability of GIS for multi-source data management and integration [10–12], while others have utilized built-in remote sensing tools to extract spatial information from microscopic image data [13–18]. Additional studies have developed methods for linking and reorientation of petrographic thin sections to real-world geographic coordinates [4,19,20]. In this respect, the work of Linzmeier et al. [21] is particularly informative, creating a framework for spatial registration of multi-source microstructural data from within a single thin section in arbitrary, two-dimensional space. By using GIS software to integrate raster images from optical and electron microscopes, along with vector point data from secondary ion mass spectrometry (SIMS) and electron probe microanalysis (EPMA), the authors mapped the distribution of structural and chemical characteristics across various crystal grains [21]. Another study that our method expands upon is that of Basil Tikoff et al. [4]. They propose a robust framework for defining the orientation of thin sections relative to sampled billets and the entire drill core, providing a tractable spatial registration method for use with both local and geographic coordinate systems. An application of the GIS-based remote sensing to Cathodoluminescence images of calcite vein networks, particularly using image classification techniques, is apparently absent from the geospatial discourse and stands as a novel technique in our study.

Cathodoluminescence (CL) image data analysis presents a useful application of the micro-GIS framework because, like land cover analysis with optical satellite images, one objective of CL is to quantify the vein network patterns by processing the changes in spectral values assigned to certain luminescing features. Structure from Motion (SFM) photogrammetry is an additional remote sensing technique that may be applied in microstructural analyses [22]. SFM is a low-cost and automated modeling approach based on the aggregation of multiple overlapping images from varying perspectives of an object or terrain. Though well established in the remote sensing literature as a viable technique

for UAS image analysis, recent studies demonstrate that SFM is also capable of accurately modeling surfaces at laboratory or microscopic scales [23].

The methods described in this study leverage geospatial tools to integrate various multi-scalar and -dimensional data layers, including tabular, graphical, and visual information, to produce spatially referenced results and accessible digital models of sample billets extracted from drill cores (Figure 1). Utilizing ESRI's ArcGIS software suite [24], this study establishes and evaluates micro-GIS procedures for compiling, managing, and analyzing those data using core samples from the San Andreas Fault Observatory at Depth (SAFOD). Our main objectives are three-fold: 1. Develop GIS-based methods for spatially referenced visualization and storage of various microstructural data from drill core billet samples; 2. Produce 3D models of sample billets and thin section positions within each billet, which serve as a digital record after irreversible material loss and fragmentation of the physical billets; and 3. Further examine the viability of the micro-GIS framework via creation of a semi-automated model for unsupervised classification of a multi-generation calcite vein network from CL imagery. Building upon previous innovative work in the field of 'micro-GIS', we address boundaries that constrain the spatially explicit examination of microstructural data and consider the challenges that persist high on the agenda of future studies.

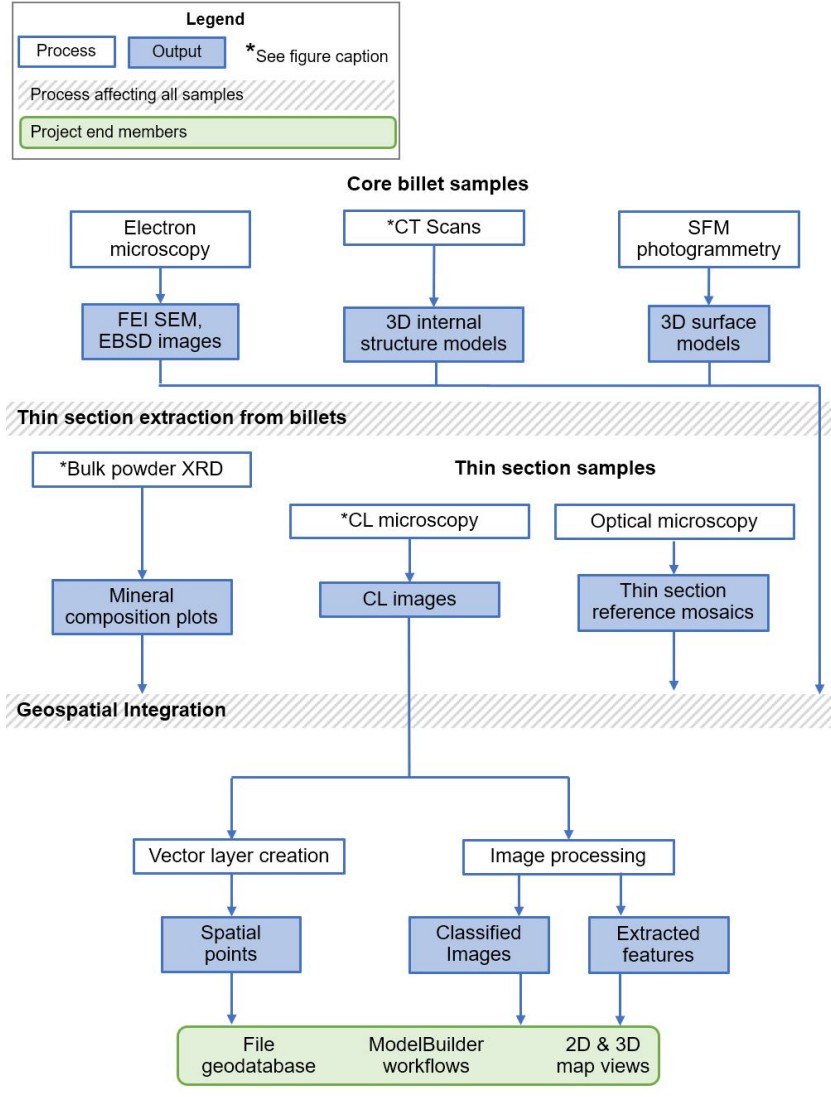

**Figure 1.** Workflow diagram of in situ data collection, processes, and resulting outputs under a GIS-based framework. Starred items indicate procedures carried out by third parties (see Acknowledgements).

## 2. Materials and Methods

### 2.1. The SAFOD Core Samples

The method for spatialized archival of core-based data in a micro-GIS environment is applied with 26 billets from the San Andreas Fault Observatory at Depth (SAFOD), which were sampled from select areas in an approximately 40 m core length during the SAFOD Phase III drilling (Figure 2). The SAFOD is located near Parkfield, California, and was drilled jointly by the National Science Foundation and the US Geological Survey beginning in 2002. Intersecting the San Andreas Fault at depths of 2–3km, the observatory provides geophysical data as well as core samples, gas, and pore-fluid samples recovered from the borehole for laboratory analysis [25].

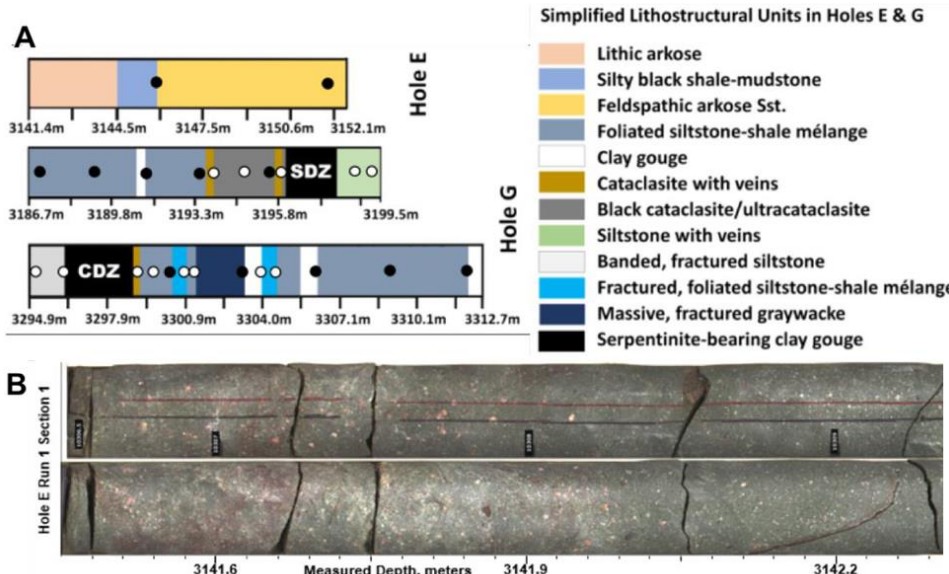

**Figure 2.** (**A**) General lithological characteristics of SAFOD core sections with a legend for units (measured depth-MD-meters). Black and white circles represent the distribution of sample billets (adapted from [26]). (**B**) Color photo of Hole E, Run 1, Section 1 including red and black core orientation lines (adapted from [27]).

The SAFOD phase III core consists of 3 intervals—Hole E, Hole G (Runs 1–3), and Hole G (Runs 4–6)—which are further subdivided into runs and sections. As in most SAFOD-related studies, numbering of the samples in this paper reflects these designations; for example, a billet extracted from Hole E, Run 1, Section 1 will be referred to as sample 'E11'. Each physical billet was marked with an arrow indicating up-borehole direction and orientation lines which establish billet rotation with respect to the core axis. Sectioning lines indicating where the billet is to be cut for thin section extraction and a 2cm scale-bar were added after the core billets were received by our laboratory.

### 2.2. Data Collection

#### 2.2.1. D Billet Models

From each billet we extracted 3D solid surface models from RGB digital camera imaging, 3D internal structure models from CT scans, and mineral composition tables from X-ray diffraction (XRD) analysis. The 3D solid surface models retain detailed surface morphology and spatial orientation markings that allow for the creation of an arbitrary local coordinate system within each billet, both of which are lost on the physical sample due to irreversible fragmentation and material loss. A Canon Powershot G1X Mark II camera with a stabilizing tripod was used to capture imagery at 12.8-megapixel resolution. The camera was placed in aperture priority (AV) mode with a wide aperture setting of F16, allowing the entire field of view to be focused. A low ISO of 200 reduced the shutter speed

and maximized the signal-to-noise ratio in the images, in turn mitigating the potential for distinct features (i.e., prominent grains and fractures) to exhibit spectral variability from one image to the next. The billets were imaged on a rotating mount inside of a controlled lighting environment.

To process the images, we used a structure for motion (SFM) photogrammetry method from the Agisoft Metashape software [28]. This procedure constructs a 3D model using the series of systematically captured, overlapping images of each billet. The slight shift in perspective between each image is exploited with an automated image alignment process that generates a 3D cloud of discrete image tie points containing color data. Ultimately, the colorized point cloud serves as the vertices from which a triangulated model surface is interpolated (Figure 3). The SFM output can then be considered as a multi-resolution 'solid surface model' consisting of a 3D object mesh bound by a photo-realistic image texture [29,30].

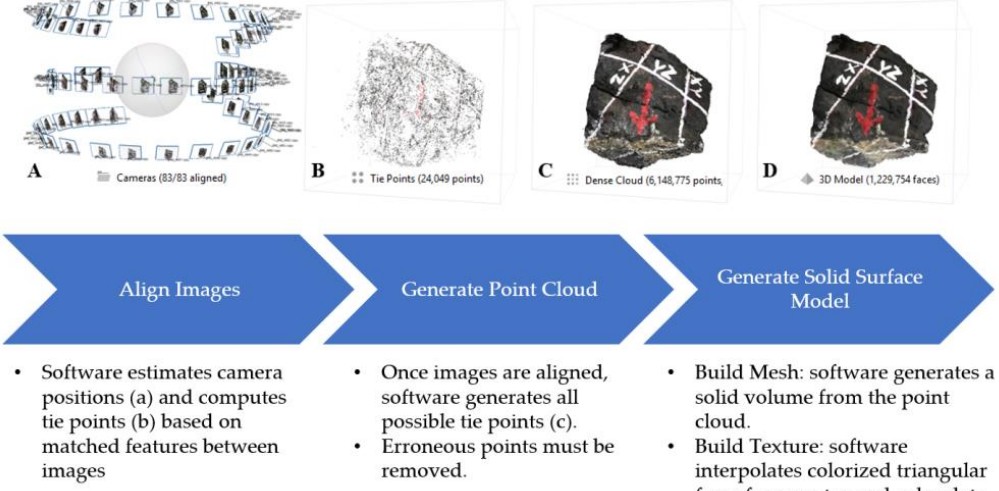

**Figure 3.** Procedure for processing 2D images to generate 3D billet models depicting: (**A**) camera positions; (**B**) initial tie points; (**C**) dense tie points cloud; and (**D**) processed 3D billet model.

The internal 3D structure models provided additional information regarding the billet interiors which were then used to identify orientation of individual billet thin sections. To extract internal 3D structures, billets were processed through a ImTek MicroCAT II CT scanner. Raw data comprises a cubic stack of DICOM images (1024 pixels$^2$ $\times$ 1024 images).

### 2.2.2. Thin Section-Based Imagery

From individual thin sections, we produced mosaics of whole sections from optical microscope images, which serve as base map layers in the GIS. Additional data include the cathodoluminescence (CL) data as well as additional image layers from the scanning electron microscope (SEM). Base maps from the thin sections were imaged using the Zeiss Axioplan optical microscope and a Scion Corporation CFW-1312C digital camera. We ensured a 30% overlap between images to ensure adequate overlap in images. Processing of the basemaps was done in the Microsoft Image Composite Editor (ICE) software [31].

### 2.3. Micro-GIS Framework

ESRI's ArcGIS v10.7.1 software served as the primary environment for integration and visualization of microstructural, compositional, and surface feature data. Its database management architecture and geospatial tools facilitated the creation of two- and three-dimensional representations of digital billet models with respect to their positions in the SAFOD core, as well as the position (or superposition) of the various layers of thin section data within the billets.

First, a new file geodatabase was created to house the data, relational information, and custom geoprocessing tool kits required for the project. Prior to importing any files, a custom arbitrary coordinate system was created and defined as geodatabase's spatial reference grid. This was a crucial step to ensure that data were archived correctly within one shared digital space. Additionally, attribute field domains and topological rules were established to enforce data integrity when importing, displaying, and editing features. Because future micro-GIS studies will encompass a multitude of approaches and data requirements, some database settings (i.e., spatial units, precision, display, and hardware utilization) utilized in this study are not reviewed in exhaustive detail.

The second step involved compiling the data into the geodatabase. This process was facilitated by initially creating an empty feature dataset corresponding to each of the data sources described throughout the paper, allowing them to be imported in batch and with respect to the defined workspace settings. The 3D solid surface models were loaded into a single 'multipatch' feature dataset. The ArcGIS-supported multipatch format reads the vertex, edge, face, and color data contained within the input COLLADA files and reconstructs an identical version of the model. 2D imagery from petrographic thin sections and sliced CT data were stored as raster datasets.

Depending on their nature and purpose, tabulated data were either imported into the geodatabase as stand-alone tables that could be graphically displayed or associated with other features via relationship classes or joined to the attributes of existing map layers if they contained direct spatial information. The XRD mineral composition plots, for example, were uploaded as stand-alone figures because their spatiality is tied to the whole extent of the corresponding billet, while the CL data table discussed later is tied directly to precise X, Y image coordinates and was joined to point features at those locations.

Using arbitrary local coordinate systems defined within individual samples, the digital billet models and corresponding thin section-based data could be stored and displayed in 3D space. Given that a reliable measured depth within a core section and the core-top direction for each sample billet is known, it was possible to define the spatial relationships of different samples in a core section using both foliation and distance. This procedure entailed assigning the centroids of the solid surface models to the corresponding placement point at the correct measured depth in local coordinates. Proceeding initial placement, the digital models were scaled using the 2 cm reference marked on the physical billet prior to imaging. The additional arrow markings were used to establish each billet's unique rotational orientation with respect to the long axis of the borehole. The up-borehole direction indicated by the arrows is, in other terms, the direction along the long axis in which measured depth in the borehole decreases.

Subsequently, the 3D surface models allowed the planar orientation of the petrographic thin sections to be defined within the local coordinates of each billet through a visual identification procedure. The planes from which the thin sections were extracted are identifiable in the 'intact' surface models by reference lines labeled on the physical billets prior to sectioning. These sectioning lines appear in photo-realistic color in the solid surface models and are made apparent in the internal structure models by placing elastic bands around the physical billets prior to CT-scanning. Because the physical cut line was visible in both models, the location of the thin section base maps and the corresponding slice of CT data could be approximated.

### 2.4. Cathodoluminescence (CL) Analysis

CL is commonly applied in solid earth science for investigations of growth and dissolution features in ore minerals, growth structures in fossils, cementation and diagenesis processes in sedimentary rocks, and the chemical and mechanical conditions of mineralized systems as they evolve through time [32]. Calcite veins are the cemented remnants of fluids introduced through repeated fracture-seal episodes within the host rock, each episode producing a new generation due to varying levels of trace impurities in the source fluid [33]. With well-established spectral proxies, CL allows identification of vein generations and the

relative time, depth, and fluid conditions in which they formed [34–38]. For the cathodo-luminescence analysis, we utilized a GIS-based workflow to 1. archive CL imagery and create point features containing spectral data and additional attribute information; and 2. extract spatially referenced information layers from raw spectral data via unsupervised classification of calcite vein generations within a thin section from the SAFOD core.

Spectral Data Identification

The first component of our analysis involves spatialized color sampling and wavelength determination of RGB pixels from luminescent areas in the CL image. Though CL records light emission from the visible portion of the electromagnetic spectrum (350–750 nm), it is not possible to quantitatively derive a spectral wavelength from a combination of RGB values because several RGB combinations exist per each unit wavelength in the 400 nm spectral range [39]. Spectral data acquisition from the pixel samples thus required a manual color matching procedure using the CIE 1931 RGB color space standard [40]. The image contains two distinct generations of calcite growth, which were visually distinguished by surveying the image for regions with pronounced contrast in apparent brightness and superimposing relationships between growth features. Figure 4 contains the CL image and provides an example of a region containing two calcite generations. Within each area of interest, the pixels in each generation with the greatest and least apparent brightness were marked with corresponding point features and then populated with their manually determined wavelengths.

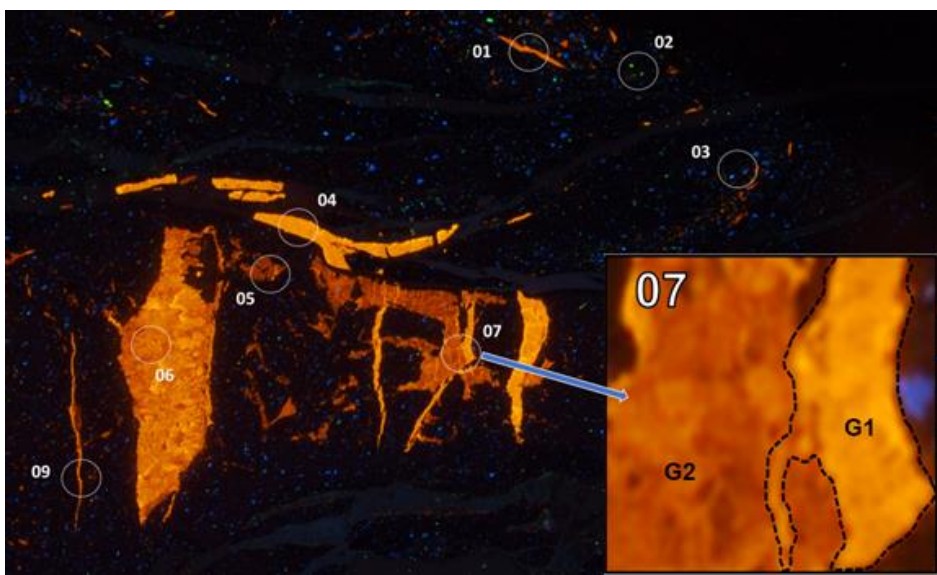

**Figure 4.** CL image containing two visually distinct generations of calcite vein growth and predefined regions of interest for spectral sampling and vector data creation. Because of significant spectral variation and superimposition among growth features, we infer that there are two distinct generations of calcite and that generation 1 (G1) is relatively younger than G2.2.4.2. CL Image Classification and Assessment.

An ISODATA unsupervised classification [41] was performed on the CL image to identify and classify calcite vein formations on the image. Input for the classification consisted of the clipped RGB raster containing only calcite, as well the FM raster (the 0–1 membership values were rescaled to the 0–255 value range of the 8-bit RGB pixels), for a total of four image bands. The classification was then run with a specified output of 10 spectral classes, which were combined as needed to achieve the final product with two apparent calcite vein generations classified.

As calcite vein generations were the sole target of classification, a fuzzy membership (FM) function was first used to isolate and extract the calcite network and remove all other image pixels from consideration. The FM function calculated the strength of pixel

membership based on the mean and standard deviation of the input RGB values (large values were specified as having high membership because luminescing calcite contains the brightest pixels in the image), generating a new raster image indicating strength of membership values ranked from 0 to 1. The membership raster was then used for two distinct purposes: (1) creating a binary dataset indicating if pixels are calcite/not calcite via a simple thresholding procedure, providing a mask for clipping the raw CL image data; and (2) input as an additional image band in the unsupervised classification algorithm (Figure 5).

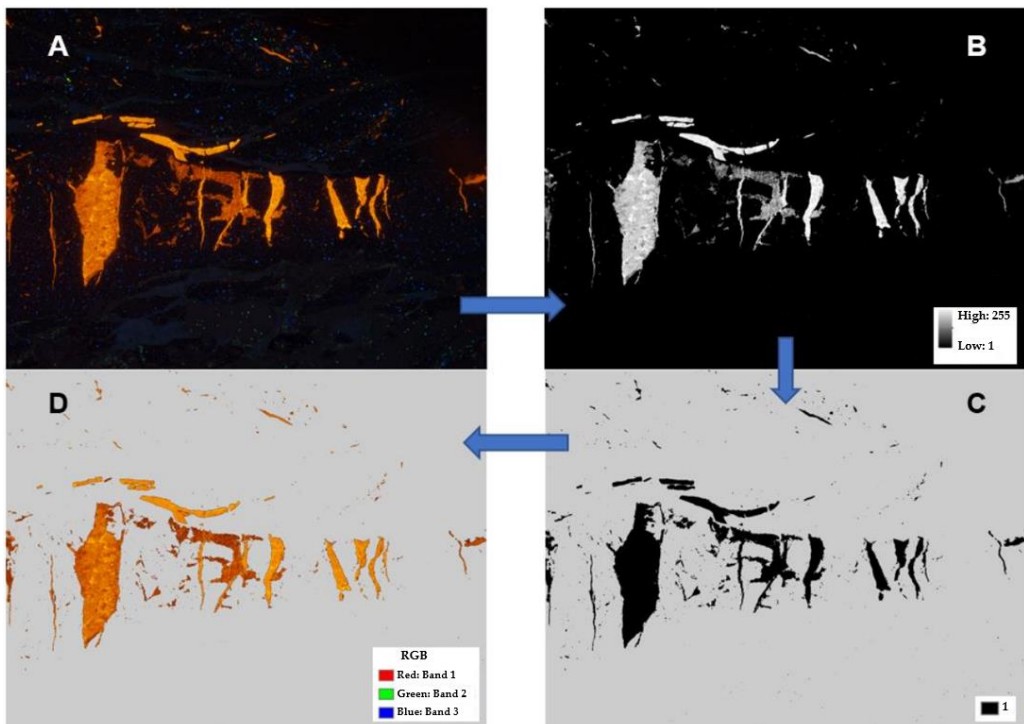

**Figure 5.** Intermediate data from the classification procedure including: (**A**) the raw CL image; (**B**) the raster band generated via the FM function, with pixel values indication strength of membership; (**C**) the binary mask band produced by thresholding the original FM image; and (**D**) the clipped raster containing RGB pixel values only within the calcite region defined by the mask band. Images B and D serve as the input for the unsupervised classification.

Accuracy assessment was conducted using 50 random points within each class (n = 100). Because a classified validation image was not available, the points were generated within the clipped raster containing only calcite and then manually assigned to the correct reference class. To reduce the potential for sampling bias, 200 points were initially produced and then reordered using randomly generated numbers. The points were then hand-classified in random order until the desired 50 points per class were obtained. A confusion matrix allowed us to quantitatively evaluate the classification performance and provided the following metrics: overall accuracy, producer's accuracy, user's accuracy, and the Kappa coefficient. Overall accuracy refers to the number of correctly classified pixels divided by the total number of pixels. Producer's accuracy is a measure of how often any one class is omitted or misclassified, while user's accuracy refers to how often within a reference class that pixels from other classes are misclassified. The Kappa coefficient is a measure of agreement between classified and reference samples and compensated for chance agreement [42]. The described workflow was implemented using the ArcGIS ModelBuilder visual programming interface, which provides a classification model framework that can be readily applied to different CL images using adjusted parameters, as needed.

## 3. Results

### 3.1. The Micro-GIS Framework

Once imported, all data could be navigated via a catalog and examined individually using the available visualization and spatial analytical tools. ArcGIS also lent many beneficial tools for establishing relationships within and between each type of data and producing visual representations that effectively conveyed meaningful information. In this study, several of such representations were produced not only to display data but also to provide means for interactive navigation throughout the available information content for all billets.

Navigation begins at the top-most spatial level of the database with a SAFOD core overview map. This map represents the core sections as polygon features and identifies each billet sample location with corresponding points, providing a simplified menu from which additional information can be explored and basic spatial inferences can be made. Arbitrary coordinates for these features were established by adopting their known measured depths as the y-axis (the x-axis was only used to define the width of the core polygons). When a point feature corresponding to a given billet is selected, the user is presented with an HTML pop-up window containing basic attribute information. The pop-up window also directly displays the mineral composition plots derived through XRD analysis, which can simply be viewed in the window or downloaded if required. Additionally, any data associated with a given billet may be accessed through hyperlinks in the pop-up (Figure 6a,b).

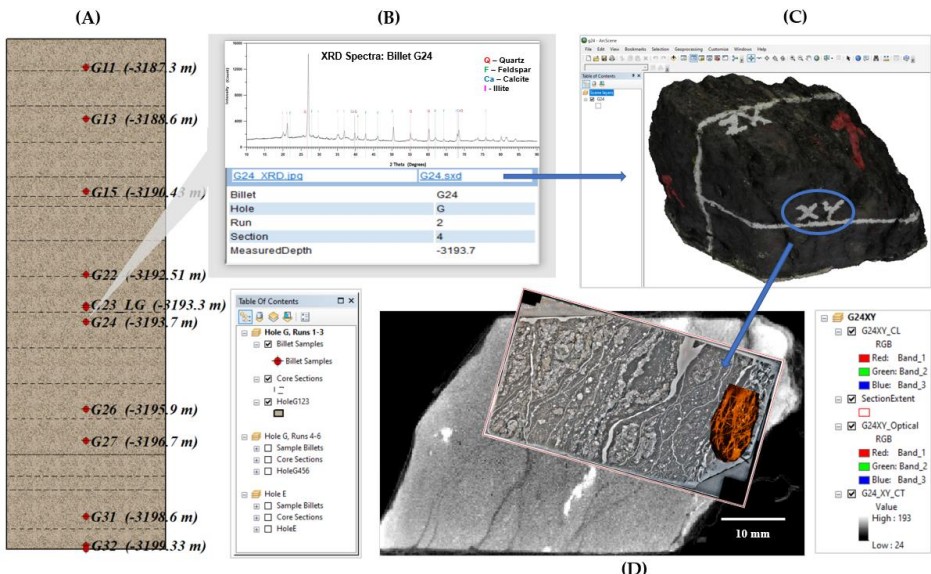

**Figure 6.** Flow diagram illustrating how project microstructural data are structured, visualized, and explored interactively within the micro-GIS. The figure depicts (**A**) the SAFOD core overview map showing sample billet locations and attributes; (**B**) An example of the HTML pop-up window containing attribute information, results from XRD analysis, and links to additional data associated with sample billet G24; (**C**) the ArcScene workspace containing billet G24′s 3D solid surface model; and (**D**) the spatially referenced 2D image data from a thin section extracted from the XY plane in G24.

The 3D solid surface models are also accessible via hyperlink in the pop-up menu or opened from direct access in the database catalog. Doing so opens an ArcScene workspace with the pre-loaded billet model, where the model may be viewed, edited, or subjected to various spatial analyses (Figure 6c). Though the whole 3D internal structure models could not be integrated directly into the database management framework, they could be accessed using hyperlinks within the GIS by providing the path to their 3DSlicer program workspace.

Pop-up windows were also enabled for the 3D billet models, allowing rapid navigation to the 2D thin section-based data acquired from within each billet (though this review follows a top-down sequence, note that links were established so that users can explore the interactive menu in any order). Organized on a per-sample basis, the thin section maps contain spatially referenced optical base mosaics, raster slices extracted from along sectioning planes in the billets' internal CT data, and CL images from SEM. The base mosaics establish primary local coordinate systems in which the other map layers are spatially registered.

### 3.2. CL Image Analysis

The results of the CL image analysis are discussed in two distinct components: (1) creating and editing of vector point features to facilitate spatialized acquisition of spectral samples; and (2) leveraging geoprocessing tools to develop a streamlined approach for calcite feature extraction, unsupervised classification of vein generations within the extracted calcite, and the presentation of accuracy metrics for f the classified output. We produced spatial features containing spectral data and their corresponding point locations in the local coordinates of the CL image. The new point features contain XY information, indication of if the sample pertains to calcite or a different luminescent mineral phase, the sample's associated calcite vein generation, and the wavelength derived by matching pixel color swatches to the CIE 1931 color standard (Figure 7). All listed attributes are of significant interest in various facets of CL-based research but primarily serve in this paper as a validation dataset for accuracy assessment of the unsupervised classification output. Nonetheless, the procedures outlined for spatial color sampling encompassed many important aspects of creating, editing, and displaying vector data in the micro-GIS environment.

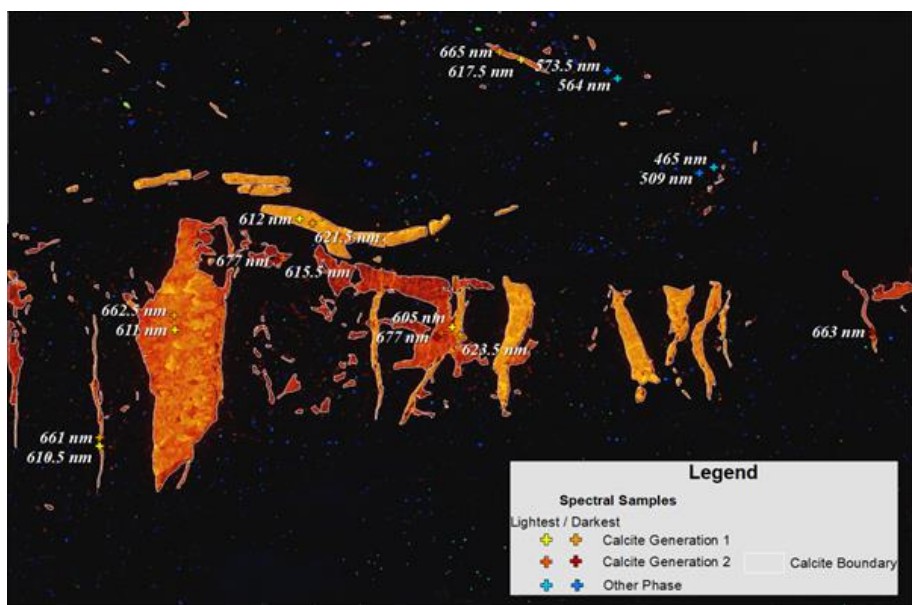

**Figure 7.** ArcGIS map view of sample points within the local coordinates of the CL image, labeled with custom symbology and relevant attribute information regarding spectral wavelength and calcite generation association.

The unsupervised classification process generated two products: the classified image indicating each pixel's membership in generation 1, 2, or neither and the transferrable model developed within ArcGIS ModelBuilder to accomplish the procedures. The procedure also generated an additional classification image band derived via the fuzzy membership (FM) function, a binary mask created by thresholding the FM raster, and the image containing only the calcite extracted by applying the mask.

The FM-derived mask preliminarily classified a total of 318,156 pixels, or approximately 6.5% of the total image, as calcite. By masking the remainder of the image, a significant amount of noise was removed from consideration by the spectral clustering algorithm, resulting in better model performance. Using both the FM raster and the clipped RGB CL image as input bands in the ISODATA classification was also found to produce more desirable spectral classes. The output of this procedure is a classified raster image that defines each pixel as either calcite generation 1, calcite generation 2, or background (Figure 8).

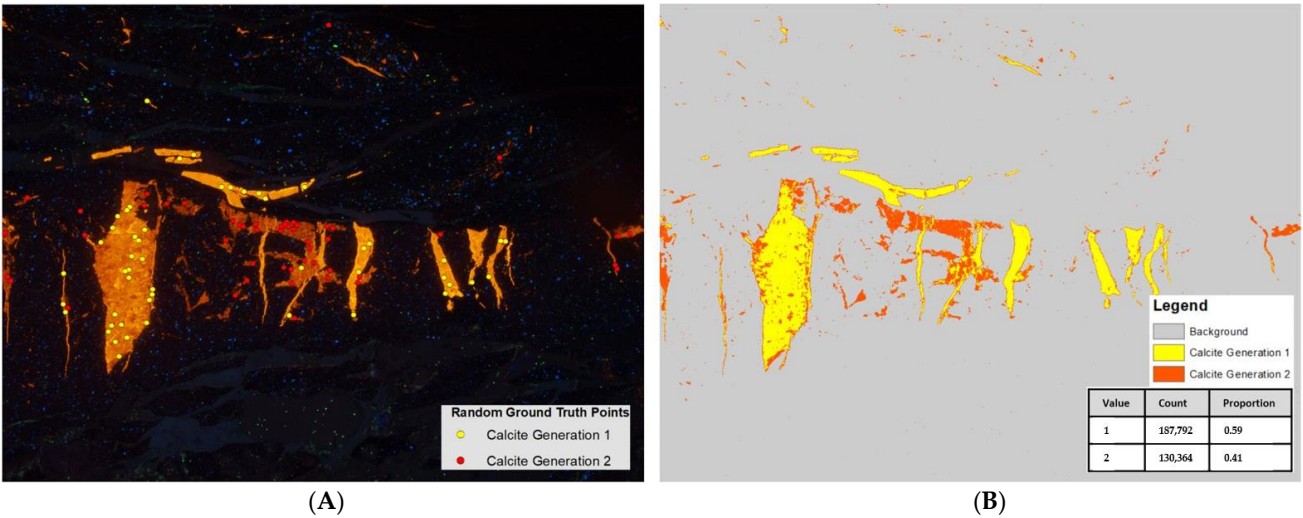

**(A)**                    **(B)**

**Figure 8.** Results of unsupervised classification showing (**A**) the input CL image, randomly generated accuracy assessment points (n = 100), and (**B**) the classified output with counts of pixels assigned as either calcite generation 1 or 2.

The results of accuracy assessment of the classified output were derived from 100 points (50 points per class) that were randomly generated within the masked calcite and assigned to the correct class. The confusion matrix indicated that overall accuracy was 85% with a Kappa coefficient of 70%. The highest producer's accuracy was observed in the generation 2 class (90%), while the highest user's accuracy (89%) was reported in the generation 1 class. These statistics indicate that of the 100 reference points, 85 were classified correctly. Of the 50 points from the generation 1 class, 20% were mis-classified as generation 2. Likewise, 10% of generation 2's points were erroneously classified as generation 1.

## 4. Discussion

Microstructural studies of fault rocks use numerous analytical and imaging techniques to conduct research, many of which produce data that contain spatially dependent information. Utilizing GIS to process these various data is beneficial because it allows them to be integrated with respect to their spatiality (1) within petrographic thin sections, (2) within core billets, and (3) within the arbitrary coordinates of the drill site and borehole. The procedures described in this paper present a universal framework for a spatially explicit management and visualization of 2D and 3D microstructural data obtained from drill core samples. Each of the various stages were ultimately marked with successes, but also identified key challenges that should be addressed in future micro-GIS efforts.

In the micro-GIS database, each 3D billet model is referenced by a corresponding placement point in a 2D overview of the SAFOD core. When a billet is selected from the overview, the user can navigate directly to the billet model and any associated microstructural data (refer to Figure 6 for an example from billet G24). Using this structure means that only the models' centroids are spatially registered and that billet rotation with respect to other billets and the core itself is not explicitly defined. In this case the exact distance between billet A's centroid and billet B's centroid (equivalent to the difference in their measured depths) can be measured, but angles or distances between any vertices in billet

A and B cannot. In future core-based GIS mapping, this study recommends that a unique spatial position for individual billets be defined within the geographic coordinates of the drill site by making full use of the trajectories recorded during drilling. Doing so would enable geometric transformations that account for the compass bearing of the core segment and the clockwise angular relationship between the billet and core orientation line. This operation would be most accurate for billets that include a portion of core's circumference; otherwise, the clockwise angle must be estimated based on the dip angle of foliation in the billet's respective section of the core (Figure 9).

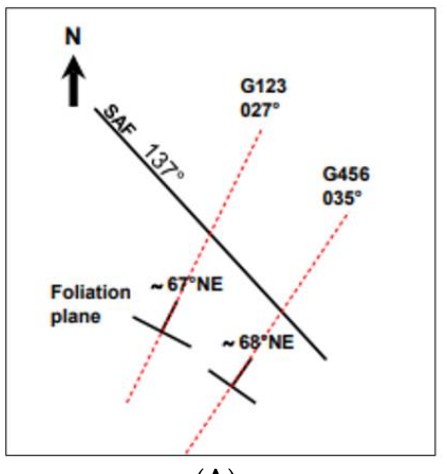
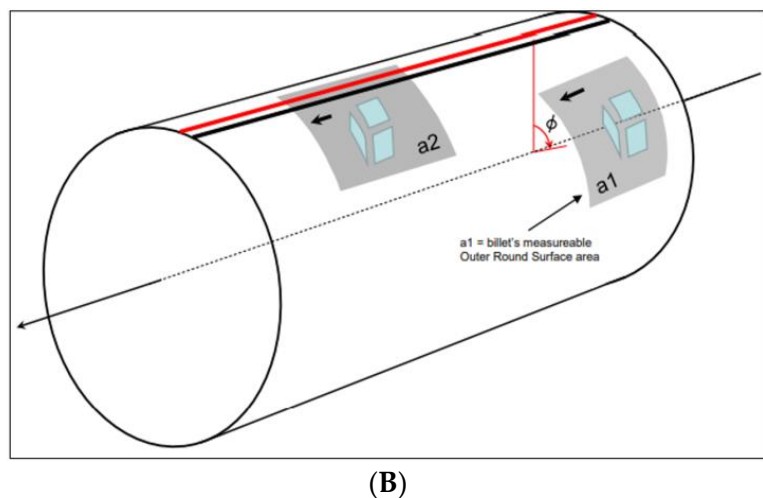

**(A)**                                                                 **(B)**

**Figure 9.** (**A**) Map view of SAF bearing 137° SE through California, core sections G123 and G456 cutting across the SAF with compass bearings of 027° NE (plunge of 67°) and 035° NE (plunge of 68°), respectively. (**B**) Schematic depiction of borehole depicting billet clockwise angle with respect to red and black core orientation lines.

The results of this study indicate that instead of tentative placement of the thin section base maps inside the models, data from thin sections could be mapped within the local 3D coordinate space of the billet model with greater precision. Each pixel in the base maps would then contain a unique XYZ position within both the billet and the borehole at-large. A recent study [4] prescribes a universal system for defining the spatial orientation of petrographic thin sections that is primarily based on a robust notching procedure that provides all information required for geometric transformation into 3D billet space (Figure 10). Though the authors did not focus on GIS-based implementation of the framework, they have suggested procedures that would complement spatially explicit map visualizations of 3D billet models and 2D thin section-based data from optical, SEM, and CL image analyses. Notably, the management of micro-structural data within a spatially referenced micro-GIS framework affords more opportunities to leverage the robust range of geospatial techniques of GIS and remote sensing. Object-based Image Analysis (OBIA) [43,44] of CL and other thin section image data represents one exciting possibility.

In this study, we employ 'traditional' pixel-based geospatial techniques of classification and raster analysis to derive meaningful information layers from the raw pixel values. This effort was greatly facilitated by the GIS database architecture, where the model could be effectively developed, tested, and modified using the ModelBuilder GUI and readily available geoprocessing tools. Furthermore, the outputs generated from the classification procedures could be seamlessly ingested as new, spatially referenced information content in the micro-GIS. As an alternative approach to the pixel-based approach in classifying the calcite vein generations, OBIA image segmentation techniques could be used to identify image objects within CL image data, and subsequent classification methods used to extract classes of features such as multiple generations of calcite veins to be stored as vector features within the micro-GIS. These vector features could then be interrogated similarly

with the micro-GIS spatial analysis techniques to characterize spatial patterns, such as trends in size, shape, density, or directional orientation.

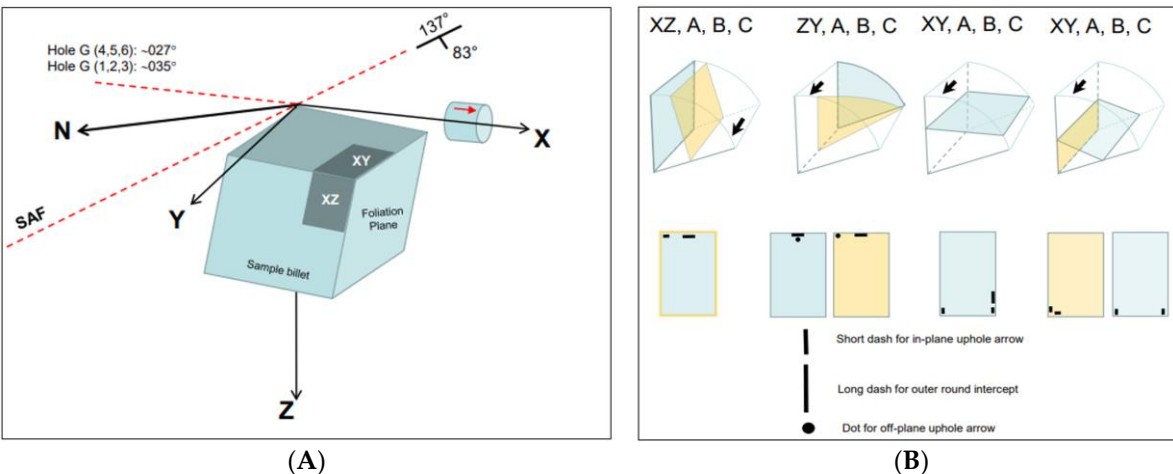

**Figure 10.** Schematic representation of thin section orientation framework depicting: (**A**) Definition of billet sectioning planes XY, XZ, and YZ with respect to within-billet foliation plane; and (**B**) Suggested markings to be placed on physical thin sections for spatial referencing within the local coordinates of billet models (2D section markings in B were adapted after Tickoff et al. [4]).

Lastly, though spatial relationships were established to some extent between all data in this study (i.e., the measured depth of the corresponding billet), the use of multiple arbitrary coordinate systems constrained the ability to achieve high spatial precision between billets and also between data products from two or more thin sections within individual billets. Initial efforts were made to establish all billet models within a single local coordinate system representing the dimensions of the SAFOD core, but were hindered by the complex transformations involved in establishing proper billet orientation with respect to the core. Similar complexities were encountered in attempts to transform planar thin section data into the 3D coordinate space of the solid surface models. The potential extensions to the spatial registration framework discussed here could possibly enhance efforts to create geographically translated forms of within-billet and -thin section coordinate systems. As such, this study suggests that improved affine transformation matrices should be developed in future research, with emphasis placed on how they might be more seamlessly integrated into the database schema to better facilitate spatial data conversion from local to geographic coordinates.

## 5. Conclusions

Utilizing ESRI's ArcGIS software suite, this study establishes micro-GIS procedures and tests the data management process with core samples from the San Andreas Fault Observatory at Depth (SAFOD). In addition to applying spatial analytical techniques on individual data layers extracted from the SAFOD thin sections, the use of a GIS framework in a microspatial context allows even further insights to be drawn. Specifically, the micro-GIS serves as an ideal repository and analytical workspace for drill core-based microstructural data that are traditionally compartmentalized. The various microstructural data (recounted below) could be imported and then managed within an optimized environment with pre-defined attribute domains, topological constraints, and custom coordinate systems suited for analyses in arbitrary space at centimeter to sub-millimeter scales. These are as follows:

1.  3D solid surface models, which provide a geometrically accurate, photorealistic representation of the physical billets;
2.  Image slices from 3D internal structure models, which consist of gridded CT data that identify internal characteristics along the sectioning planes of the billets;

3. X-ray diffraction (XRD) histogram plots that characterize the mineralogical composition of the billets;
4. Thin section optical mosaics that serve as a base map layer from which the spatial positions of other thin section-based data can be registered; and
5. SEM-CL image data which contain high-resolution spectral information from areas of interest within the thin sections.

This study sought to probe the potential for treatment of those data with the geospatial framework in a study involving 26 sample billets from the SAFOD. Multiple forms of 3D and 2D data were integrated in a geodatabase that archived not only the input data, but also their later-defined spatial relationships and outputs derived from spatial analyses. By-products of the method included interactive map visualizations, a hierarchical data catalog based on spatial relationships, and ModelBuilder workflows for accomplishing the various database management and analysis tasks.

The overarching concept inherent to GIS is that of relating different, spatially overlapping data to interrogate associations between data types. If data are adequately spatially referenced, it has the potential to assess relationships between data sources that may motivate further research and methods development focused on the micro-GIS framework. Novelty may be found specific to micro-GIS analyses for characterizing a host of deformation microstructures including complex, multi-generation networks of calcite and quartz veins, as preliminarily explored in the CL analysis of this paper.

To this end, future structural geology studies should continue prioritizing the spatial relationships within and between the different sources of microstructural data, but also work toward a more standardized vernacular around spatially explicit handling of those data in a GIS framework. For example, civil engineers benefit from tailored analytical toolkits such as 'CityEngine' for 3D modeling and 'Network Analyst' for studies involving infrastructure and transportation data, while hydrologists have access to specific tools for delineating watersheds, estimating the flow and accumulation of surface water, and modeling the path of groundwater contaminants [24]. In a similar fashion, custom geoprocessing tools and workspace templates should also be developed to better facilitate structural geologists working at the micro-scale.

Our study confronted the procedural and conceptual challenges associated with spatial integration of core-based microstructural data, built on recent advances to develop universal micro-GIS procedures, and demonstrated the usefulness of GIS within the broader context of structural geology as a whole. With continued advances on both disciplinary fronts, the novel approaches discussed in this study are well-positioned to inform, and simultaneously be informed by, innovations in the future.

**Author Contributions:** Conceptualization: Elliott M. Holmes, Andrea E. Gaughan, Donald J. Biddle, Forrest R. Stevens, and Jafar Hadizadeh; Methodology: Elliott M. Holmes, Andrea E. Gaughan, Donald J. Biddle, and Forrest R. Stevens; Data Curation: Elliott M. Holmes and Jafar Hadizadeh; Writing—Original Draft Preparation: Elliott M. Holmes; Writing—Review and Editing: Elliott M. Holmes, Andrea E. Gaughan, Donald J. Biddle, Forrest R. Stevens, and Jafar Hadizadeh; Figure Editing: Elliott M. Holmes; Supervision: Andrea E. Gaughan and Jafar Hadizadeh; Funding Acquisition: Jafar Hadizadeh, Andrea E. Gaughan, and Donald J. Biddle. All authors have read and agreed to the published version of the manuscript.

**Funding:** This research was partially supported by the U.S. National Science Foundation grant NSF-EAR-1800933 to J. Hadizadeh, A.E. Gaughan, and D.J. Biddle.

**Institutional Review Board Statement:** Not applicable.

**Informed Consent Statement:** Not applicable.

**Data Availability Statement:** The data presented in this study are available on request from the corresponding author. The data are primarily derived from microstructural and geospatial analyses and are not publicly available due to large file dimensions.

**Acknowledgments:** The authors express our appreciation for the data curation efforts of Huaiyu Zheng, Aryan Ghazipour, and Grace Embree (CT scans), in addition to Alan Boyle (CL imaging). We thank Judith Chester, Texas A&M University, chair of the NSF SAFOD sample committee for providing the core samples.

**Conflicts of Interest:** The authors declare no conflict of interest.

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
