# Peer review of "Geospatial Management and Analysis of Microstructural Data from San Andreas Fault Observatory at Depth (SAFOD) Core Samples"

_ijgi, doi:10.3390/ijgi10050332_

Round 1

Reviewer 1 Report

Dear Authors,

The current manuscript needs your attention in several parts for minor modifications. As general comments, the manuscript addresses an interesting topic of Geological characterization and remote sensing in micro scale.

You can find specific comments and suggestions in the attached file.

Author Response

Dear Reviewer,

We that you for your careful review of our manuscript. Your suggestions have improved the original manuscript, making it more clear and cogent for the reader. These revisions include improvements to paragraph/sentence structure, replacing blurry figures with sharper images, and several typographical changes. A detailed, point-by-point response to all feedback can be found below:

  1. Line 17- Ensure that CL abbreviation is identified as cathodoluminescence.
    • Corrected to read “…cathodoluminescence (CL) imagery.”
  2. Line 93- Include the term photogrammetry when introducing SFM.
    • Original text: “In addition, another remote sensing technique that translates to microstructural analysis is Structure for motion (SFM) photogrammetry [22].”
    • Revised text: “Structure from Motion (SFM) photogrammetry is an additional remote sensing technique that may be applied in microstructural analyses [22}.”
  3. Line 96- Remove term “2D” and add term “overlapping” when describing image capture.
    • Original text: “…multiple 2D images…”
    • Revised text: “…multiple overlapping images…”
  4. Line 101- Explain more explicitly the nature of project data.
    • Original text: “The methods described in this study leverage GIS tools to integrate various multi-dimensional data layers, both numerical and visual, to produce accurately referenced results and digital models of sample billets extracted from drill cores. Utilizing ESRI’s ArcGIS software suite [24], this study establishes micro-GIS procedures and tests the data management process with core samples from the San Andreas Fault Observatory at Depth (SAFOD).”
    • Revised text: “The methods described in this study leverage geospatial tools to integrate various multi-scalar and -dimensional data layers, including tabular, graphical, and visual information, to produce spatially referenced results and accessible digital models of sample billets extracted from drill cores. Utilizing ESRI’s ArcGIS software suite [24], this study establishes and evaluates micro-GIS procedures for compiling, managing, and analyzing those data using core samples from the San Andreas Fault Observatory at Depth (SAFOD).”
  5. Figure 1- Change “Micro-GIS Demonstration” to “Geospatial Integration”.
    • Revised as requested.
  6. Figures 2 & 3- Replace blurred figures with images of higher quality.
    • Revised as requested.
  7. Line 150- Provide camera resolution and specifications.
    • Original text: “Imaging was done using a Canon Powershot G1X Mark II camera with a stabilizing tripod, while the billets were placed on a rotating mount inside of a controlled lighting environment.”
    • Revised text: “A Canon Powershot G1X Mark II camera with a stabilizing tripod was used to capture imagery at 12.8-megapixel resolution. The camera was placedin aperture priority (AV) mode with a wide aperture setting of F16, allowing the entire field of view to be focused. A low ISO of 200 reduced the shutter speed and maximized the signal-to-noise ratio in the images, in turn mitigating the potential for distinct features (i.e. prominent grains and fractures) to exhibit spectral variability from one image to the next. The billets were imaged on a rotating mount inside of a controlled lighting environment.”
  8. Line 189- Identified as unnecessary information.
    • Removed text: “To load and work with the various modes of data in the GIS, the following were considered.”
  9. Line 195- Not clear for the reader.
    • Original text: “Other miscellaneous geodatabase settings were adjusted to suit more specific needs of certain data but are not described in extraneous detail due the multitude of approaches that could be utilized in other micro-GIS endeavors.”
    • Revised text: “Because future micro-GIS studies will encompass a multitude of approaches and data requirements, some database settings (i.e. spatial units, precision, display, and hardware utilization) utilized in this study are not reviewed in exhaustive detail.”
  10. Line 221- Not clear for the reader.
    • Original text: “The arrow indicating the up-borehole direction then allowed the models to be oriented relative to the long axis of the core.”
    • Revised text: “The additional arrow markings were used to establish each billet’s unique rotational orientation with respect to the long axis of the borehole. The up-borehole direction indicated by the arrows is, in other terms, the direction along the long axis in which measured depth in the borehole decreases.”
  11. Line 258 (Figure 4)- Should be explained better in the text.
    • Original text: “The image contains two generations of calcite, distinguishable by significant luminescence contrast and cross-cutting shape relationships (Figure 4). Within each area of interest, the lightest and darkest pixel of each generation was identified with a corresponding point feature and then populated with its manually determined wavelength.”
    • Revised text: “The image contains two distinct generations of calcite growth, which were visually distinguished by surveying the image for regions with pronounced contrast in apparent brightness and superimposing relationships between growth features. Figure 4 contains the CL image and provides an example of a region containing two calcite generations. Within each area of interest, the pixels in each generation with the greatest and least apparent brightness were marked with corresponding point features and then populated with their manually determined wavelengths.
      1. Revised Figure 4 caption: “CL image containing two visually distinct generations of calcite vein growth and predefined regions of interest for spectral sampling and vector data creation. Because of significant spectral variation and superimposition among growth features, we infer that there are two distinct generations of calcite and that generation 1 (G1) is relatively younger than G2.”
  1. Line 299- Please restructure the paragraph.
    • Original text: “A confusion matrix was then provided the following metrics: overall accuracy- the number of correctly classified pixels divided by the total number of pixels; producer’s accuracy- how often any one class is omitted or misclassified; user’s accuracy- how often within a reference class are pixels from other classes misclassified; and the Kappa coefficient agreement between classified and reference samples compensated for chance agreement [42].”
    • Revised text: “A confusion matrix allowed us to quantitatively evaluate the classification performance and provided the following metrics: overall accuracy, producer’s accuracy, user’s accuracy, and the Kappa coefficient. Overall accuracy refers to the number of correctly classified pixels divided by the total number of pixels. Producer’s accuracy is a measure of how often any one class is omitted or misclassified, while user’s accuracy refers to how often within a reference class that pixels from other classes are misclassified. The Kappa coefficient is a measure of agreement between classified and reference samples and compensated for chance agreement [42].”
  2. Figure 8- Proportion decimals should be limited.
    • Limited to 2 decimal places.
  3. Line 413- The current paragraph needs revision. It is confusing for the reader in several parts and sentences.
    • Original text: “In the micro-GIS database, the 3D billet models are tied by their centroids to a corresponding placement point at the correct measured depth in a 2D SAFOD core overview map. In other words, the models’ centroids are the only precise locations that spatially distinguish the models from one another. In this case the exact distance between billet A’s centroid and billet B’s centroid can be measured, but angles or distances between any two vertices in billet A and B cannot. In future core-based GIS mapping, this study recommends a unique spatial position for individual billets be defined within the geographic coordinates of the drill site by making full use of the geophysical logs recorded during drilling. Doing so would enable geometric transformations that account for the compass bearing of the core segment and the clockwise angular relationship between the billet and core orientation line. This operation would be most accurate for billets that include a portion of the outer round surface of the core; otherwise, the clockwise angle must be estimated based on the dip of the foliation plane in the billet’s respective core section (Figure 9).”
    • Revised text: “In the micro-GIS database, each 3D billet model is referenced by a corresponding placement point in a 2D overview of the SAFOD core. When a billet is selected from the overview, the user can navigate directly to the billet model and any associated microstructural data (refer to Figure 6 for an example from billet G24). Using this structure means that only the models’ centroids are spatially registered and that billet rotation with respect to other billets and the core itself is not explicitly defined. In this case the exact distance between billet A’s centroid and billet B’s centroid (equivalent to the difference in their measured depths) can be measured, but angles or distances between any vertices in billet A and B cannot. In future core-based GIS mapping, this study recommends that a unique spatial position for individual billets be defined within the geographic coordinates of the drill site by making full use of the trajectories recorded during drilling. Doing so would enable geometric transformations that account for the compass bearing of the core segment and the clockwise angular relationship between the billet and core orientation line. This operation would be most accurate for billets that include a portion of core’s circumference; otherwise, the clockwise angle must be estimated based on the dip angle of foliation in the billet’s respective section of the core (Figure 9).”
  4. Line 478- Repetitive phrase.
    • Removed text: “Our study saw the successful development of a micro-GIS that facilitated efforts to integrate, visualize, and examine the data content of the SAFOD sample billets.”
  5. Line 511- Already done in literature.
    • Original text: “Another case can be observed with hydrology-specific tools for identifying watersheds, estimating the flow and accumulation of surface water, and modelling the path of groundwater contaminants.”
    • Author’s Note: We intended to highlight the hydrology tools as already well-established in the literature, much like the civil engineering tools in the sentence prior. We have attempted to clarify this with the revised text.
      1. Revised text: “For example, civil engineers benefit from tailored analytical toolkits such as ‘CityEngine’ for 3D modelling and ‘Network Analyst’ for studies involving infrastructure and transportation data, while hydrologists have access to specific tools for delineating watersheds, estimating the flow and accumulation of surface water, and modelling the path of groundwater contaminants [24].”
  1. Several Instances- Erroneous double space after sentences.
    • Revised as requested.

Reviewer 2 Report

The structure of the article is well developed, and the topic is very interesting because shows in an exhaustive way a new GIS-based approach in an unusual field of research, opening new scenarios for micro-GIS procedures.

I suggest only few modifications to the paper

line 62 insert GNSS instead of GPS

line 62-63 please explicit the concept in a clearer way

Figure 1. try to make the schema more clear (the arrow that starts from "CL images"
is longer than the others)

line 153-154. please explicit the concept in a clearer way

line 180. please introduce the workflow in a better form.

line 371. "of the image" is repeated.

line 412. please explicit the concept in a clearer way

Author Response

Dear Reviewer,

The authors express our appreciation for your interest in our topic and your suggestions that have led to improvement of the submitted manuscript. We accept your revisions and have detailed below how the feedback was implemented into the study:

  1. Line 62- Insert GNSS instead of GPS, explain concept more clearly.
    • Original text: “Studies typically span scales of meters to kilometers and may integrate multiple sources of remotely sensed aerial imagery and vector data acquired with GPS-enabled devices. One unifying aspect among these applications is the importance of the coordinate space that defines the study area; in other terms, the generated results are often inseparable from their spatial context.
    • Revised text: “Studies typically span scales of meters to kilometers and may integrate multiple sources of remotely sensed aerial imagery and vector data acquired with GNSS-enabled devices. One unifying aspect among these applications is the importance of the coordinate space that defines the study area; in other terms, the spatial context of research results is often of equal importance to, or inseparable from, the research itself.”
  2. Figure 1- Clarify Schema.
    • Figure schema adjusted to clarify project workflow.
  3. Line 159- Explain concept more clearly.
    • Original text: “The general procedure involves initial photo alignment that generates a 3D cloud of discrete image tie points containing color data, which ultimately serve as the vertices from which triangular faces are interpolated in the final model (Figure 3).”
    • Revised text: “This procedure constructs a 3D model using the series of systematically captured, overlapping images of each billet. The slight shift in perspective between each image is exploited with an automated image alignment process that generates a 3D cloud of discrete image tie points containing color data. Ultimately, the colorized point cloud serves as the vertices from which a triangulated model surface is interpolated (Figure 3).”
  4. Line 189- Introduce the workflow in a better form.
    • Original text: “To load and work with the various modes of data in the GIS, the following were considered. First, a new file geodatabase was created to house the data, relational information, and custom geoprocessing tool kits required for the project.”
    • Reviewer 1 asked that the first sentence simply be removed.
      1. Revised text: “First, a new file geodatabase was created to house the data, relational information, and custom geoprocessing tool kits required for the project.”
  1. Line 384- “of the image” is repeated.
    • Removed duplicate text.
  2. Line 413- Explain concept more clearly.
    • Original text: “In the micro-GIS database, the 3D billet models are tied by their centroids to a corresponding placement point at the correct measured depth in a 2D SAFOD core overview map. In other words, the models’ centroids are the only precise locations that spatially distinguish the models from one another. In this case the exact distance between billet A’s centroid and billet B’s centroid can be measured, but angles or distances between any two vertices in billet A and B cannot. In future core-based GIS mapping, this study recommends a unique spatial position for individual billets be defined within the geographic coordinates of the drill site by making full use of the geophysical logs recorded during drilling. Doing so would enable geometric transformations that account for the compass bearing of the core segment and the clockwise angular relationship between the billet and core orientation line. This operation would be most accurate for billets that include a portion of the outer round surface of the core; otherwise, the clockwise angle must be estimated based on the dip of the foliation plane in the billet’s respective core section (Figure 9).”
    • Revised text: “In the micro-GIS database, each 3D billet model is referenced by a corresponding placement point in a 2D overview of the SAFOD core. When a billet is selected from the overview, the user can navigate directly to the billet model and any associated microstructural data (refer to Figure 6 for an example from billet G24). Using this structure means that only the models’ centroids are spatially registered and that billet rotation with respect to other billets and the core itself is not explicitly defined. In this case the exact distance between billet A’s centroid and billet B’s centroid (equivalent to the difference in their measured depths) can be measured, but angles or distances between any vertices in billet A and B cannot. In future core-based GIS mapping, this study recommends that a unique spatial position for individual billets be defined within the geographic coordinates of the drill site by making full use of the trajectories recorded during drilling. Doing so would enable geometric transformations that account for the compass bearing of the core segment and the clockwise angular relationship between the billet and core orientation line. This operation would be most accurate for billets that include a portion of core’s circumference; otherwise, the clockwise angle must be estimated based on the dip angle of foliation in the billet’s respective section of the core (Figure 9).”

Reviewer 3 Report

The paper contains no discussion how standard geospatial analysis techniques apply to micro-GIS scenario and this specific use-case. The authors should illustrate usability of micro-GIS in this way. 

It would also be interesting to present what geospatial analysis techniques would be specific to micro-GIS that otherwise did not exist in traditional GIS.

Section 2.2 Data collection describes in too much details (5 out of 15 pages of the paper) process of acquiring data which is not of primary interest to readers of IJGI. This section should probably be shortened. Instead, authors should add discussion how micro-GIS can help analyze/visualize artefacts from data collection phase.

Author Response

Dear Reviewer,

The authors express our appreciation for your interest in our topic and careful review of the submitted manuscript. The reviews strengthen our paper and we have detailed below how the feedback was implemented into the text:

  1. The paper contains no discussion how standard geospatial analysis techniques apply to micro-GIS scenario and this specific use-case. The authors should illustrate usability of micro-GIS in this way. 
    • We thank the author for the suggestion and have added a paragraph to the discussion to reflect further geospatial techniques applied in a micro-GIS context.
    • Revised text: Notably, the management of micro-structural data within a spatially referenced micro-GIS framework affords more opportunities to leverage the robust range of geospatial techniques of GIS and remote sensing. Object-based Image Analysis (OBIA) [43, 44] of CL and other thin section image data represents one exciting possibility. In this study, we employ “traditional” pixel-based geospatial techniques of classification and raster analysis to derive meaningful information layers from the raw pixel values. This effort was greatly facilitated by the GIS database architecture, where the model could be effectively developed, tested, and modified using the ModelBuilder GUI and readily available geoprocessing tools. Furthermore, the outputs generated from the classification procedures could be seamlessly ingested as new, spatially referenced information content in the micro-GIS. As an alternative approach to the pixel-based approach in classifying the calcite vein generations, OBIA image segmentation techniques could be used to identify image objects within CL image data, and subsequent classification methods used to extract classes of features such as multiple generations of calcite veins to be stored as vector features within the micro-GIS. These vector features could then be interrogated similarly with the micro-GIS spatial analysis techniques to characterize spatial patterns, such as trends in size, shape, density, or directional orientation.
  2. It would also be interesting to present what geospatial analysis techniques would be specific to micro-GIS that otherwise did not exist in traditional GIS.
    • We appreciate the reflection of potentials of how geospatial analysis techniques would be specific to micro-GIS but the concepts of traditional GIS translate over to a micro-GIS context, we argue it’s just a matter of scale and proper spatial referencing that opens up many doors to spatially-explicit interrogation of microstructural data. We have added a couple sentences in the conclusion to reflect this and the potential of micro-GIS.
    • Revised text: The overarching concept inherent to GIS is that of relating different, spatially overlapping data to interrogate associations between data types. If data are adequately spatially referenced it is the potential to assess relationships between data sources that may motivate further research and methods development focused on the micro-GIS framework. Novelty may be found specific to micro-GIS analyses for characterizing a host of deformation microstructures including complex, multi-generation networks of calcite and quartz veins, as preliminarily explored in the CL analysis of this paper.
  3. Section 2.2 Data collection describes in too much detail (5 out of 15 pages of the paper) process of acquiring data which is not of primary interest to readers of IJGI. This section should probably be shortened. Instead, authors should add discussion how micro-GIS can help analyze/visualize artefacts from data collection phase.
    • Section 2.2 is just over 1 page so we are not sure where the reviewer would like to see methods reduced. We also argue that the detail in methods is important for this type of paper where readers will be interested in the construction of the micro-GIS and data compiled for the database. With that said, we went back and edited the text throughout Section 2 while still addressing other reviewer comments and retaining the detail of the fundamental workflow.

Round 2

Reviewer 3 Report

The authors have appropriatelly responded to reviewers' comments and modified the paper accordingly.